# CONDITIONAL EXECUTION OF CASCADED MODELS IMPROVES THE ACCURACY-EFFICIENCY TRADE-OFF

## ABSTRACT

The compute effort required to perform inference on state-of-the-art deep learning models is ever growing. Practical applications are commonly limited to a certain cost per inference. Cascades of pretrained models with conditional execution address these requirements based on the intuition that some inputs are easy enough that they can be processed correctly by a small model allowing for an early exit. If the small model is not sufficiently confident in its prediction, the input is passed on to a larger model. The selection of the confidence threshold allows to trade off compute effort against accuracy. In this work, we explore the effective design of model cascades, and thoroughly evaluate the impact on the accuracy-compute trade-off. We find that they not only interpolate favorably between pretrained models, but that this trade-off curve commonly outperforms single models. This allows us to redefine most of the ImageNet Pareto front already with 2-model cascades, achieving an average reduction in compute effort at equal accuracy of almost $3.1\times$ above 86% and more than $1.9\times$ between 80% and 86% top-1 accuracy. We confirm the wide applicability and effectiveness of the method on the GLUE benchmark. We release the code to reproduce our experiments in the supplementary material and use only publicly available models and datasets.

## 1 INTRODUCTION

The trade-off between accuracy and efficiency is fundamentally important to deep learning. While state-of-the-art results are achieved by models of ever growing size, practical applications are constrained. Reducing the energy consumption is important for energy limited systems like wearable and mobile devices. For inference at large scale such as in data centers, minimizing the compute resource requirements is important economically and environmentally.

There are many different approaches to improve the accuracy-efficiency trade-off (see Section 2). We explore early exiting from cascades of pretrained models for classification tasks. Such an approach can work when some inputs are easier to classify than others and therefore can be processed by a smaller model to save computation. Assuming the input's difficulty is unknown, a first classification can be made by the smaller model and if it is not confident in its prediction, the input can be classified again by a larger and more accurate model. The models in an early exit cascade are ordered by cost from cheapest to most expensive and an early exit is made after the first model where the prediction confidence satisfies the exit condition.

Early exiting is often explored from within a model like in BranchyNet (Teerapittayanon et al., 2016) or Shallow-Deep Networks (Kaya et al., 2019), by inserting additional classifier outputs earlier in the model. Huang et al. (2017) describe how early classifiers lack coarse features and training the model for early classification can lower the accuracy of later classifiers. Furthermore, modern convolutional neural networks are highly optimized and their depth, width and resolution is carefully scaled, gradually lowering the resolution and increasing the number of channels as features become more complex (Tan & Le, 2019). For early classification outputs, this optimization and scaling is disrupted, which results in a lower accuracy than a separate model at equal compute effort can achieve. We demonstrate experimentally that a smaller accuracy difference between classifiers allows for more frequent early exiting. This motivates using a separate model for early classification, which enables us to increase the early exit rate at the cost of a small overhead when forwarding to the larger model, since features are no longer shared between classifiers. However, early exit-

ing from within a single model and between cascaded models are two separate and complementary approaches that can be combined (Bolukbasi et al., 2017).

A disadvantage of early exit model cascades is that more than one model is required, which increases memory as well as worst case latency and compute effort, where the worst case is no early exit. However, the costs are dominated by the largest model, which means that the added cost of smaller models is usually negligible. Key advantages are the ease of use and efficacy. Pretrained models can be used as is or finetuned. Cascading is a complementary approach and can be combined with many other approaches for increasing efficiency, particularly methods that improve the individual models, such as better architecture and training. When the input is processed by at least 2 models in the cascade, the predictions can be ensembled. Gontijo-Lopes et al. (2021) show that ensembles are more accurate when the models are more diverse due to the lower error correlation, which can also be leveraged by cascades.

Although early exit model cascades are simple and effective, little prior work on them has been published (see Section 2). We explore how to build such cascades effectively on ImageNet (Russakovsky et al., 2015). For this we compare many different cascading methods across continuous Pareto fronts in multiple settings using a diverse selection of pretrained models for a thorough and reliable evaluation. We then confirm their wide applicability on text classification tasks from the GLUE (Wang et al., 2018a) benchmark. Our contributions and findings are:

- We demonstrate that already 2-model cascades dominate the entire ImageNet Pareto front.
- We provide insight into how to construct the model cascades: We empirically demonstrate a relationship between accuracy difference of cascaded models and achievable early exit rate as well as desirable size difference in Figure 2.
- The maximum softmax confidence metric achieves most improvement overall while softmax margin excels in low confidence scenarios and the commonly used entropy metric performs worst.
- Ensembling predictions, when inference is done for at least 2 models in a cascade, generally increases the top accuracy achieved by the cascade but lowers Pareto improvement. Temperature scaled calibration is ineffective at alleviating this.
- For our evaluation we rely on an external baseline comprised of Pareto optimal public models to ensure our results are credible and relevant. This means we provide a strong and reproducible baseline for future related research that is currently missing.

## 2 RELATED WORK

**Accuracy-efficiency trade-off** Improving the accuracy-efficiency Pareto front is a central objective for a wide breadth of research with many different approaches. MobileNetV3 (Howard et al., 2019) and EfficientNet (Tan & Le, 2019) represent how architectures have become highly optimized. Model efficiency can be improved further by applying compression techniques like quantization and pruning (Han et al., 2015). Once-for-all (Cai et al., 2019) utilizes progressive shrinking of input resolution, kernel size, network width and depth together with knowledge distillation to obtain more efficient models than conventional neural architecture search. Training is very important with recent advances most notably through data augmentation (Shorten & Khoshgoftaar, 2019) and pretraining on more data (Ridnik et al., 2021), which is enabled further by self-supervised (Devlin et al., 2018) and semi-supervised (Pham et al., 2021) methods. Many other procedures exist such as model soups (Wortsman et al., 2022), which fine-tunes a model with multiple hyperparameter configurations and averages the weights. More closely related to our work are dynamic neural networks (Han et al., 2021; Xu & McAuley, 2022). An example for dynamic depth is SkipNet (Wang et al., 2018b), which adds gating modules and a learned skipping policy to skip network layers based on the input. BranchyNet (Teerapittayanon et al., 2016) adds branches to the original net for early evaluation and exits when prediction entropy is below a threshold. Wołczyk et al. (2021) improve early exiting from within a model by recycling predictions of earlier classifiers. We focus on early exiting from a cascade of models, which is a complementary approach that is comparatively simple yet effective.

**Early exit model cascades** Cascades are commonly used in machine learning and have been popularized by influential works such as Viola & Jones (2001). We focus on a specific type of cascade

which uses multiple models in sequence with the capability to exit early if confident enough in a prediction. Park et al. (2015) propose the big/LITTLE architecture, which is a cascade of a small and large model with static or dynamic softmax margin threshold as exit condition to reduce energy consumption. Wang et al. (2017) demonstrate reduced computational cost by cascading up to 3 models. They compare different classifiers for the objective of achieving a desired accuracy, which include a trained decision network, cost-oblivious grid search for both entropy and maximum softmax thresholds, and cost-aware learned entropy threshold. They find that only the last three methods manage to satisfy the accuracy objective, that entropy is a better confidence metric than maximum softmax and that the cost-aware method performs better. Guan et al. (2017) train an agent to exit between models to reduce computational cost. Similarly, Bolukbasi et al. (2017) present trained policies to exit early between as well as within models for a trade-off between accuracy and inference time. Streeter (2018) proposes an algorithm to automatically construct efficient cascades from a selection of available models. Inoue (2019) show that exiting early from an ensemble of 10 or 20 models can greatly reduce computational cost while preserving most of the static ensemble's accuracy. They use average softmax to ensemble the predictions and exit based on a confidence interval condition. Wang et al. (2022) cascade up to 4 models of the same architecture families such as EfficientNet (Tan & Le, 2019), ensemble by averaging the logits, use maximum softmax as confidence metric, and optimize for accuracy and computational cost with exhaustive search. They also demonstrate the efficacy of self-cascading by varying the input resolution for the same model.

From prior work, we notice a lack of comparison between different cascading methods, knowledge gaps about when cascades perform well, and limited evaluation. We address this by comparing many different cascading methods in multiple settings using a diverse selection of pretrained models for a thorough and reliable evaluation. We investigate when cascades work well and unlike prior work we evaluate improvement across continuous Pareto fronts. No training is done in our work. Instead, we use publicly available state-of-the-art models and make available code to reproduce our results, thereby providing a strong baseline for future related research that is currently missing.

## 3 METHOD

The idea behind early exit cascades is that inference can be done on a smaller model first and if confident in its prediction we can save time and computation by exiting early and skipping the original model. A cascade contains at least 2 pretrained models. A condition is needed to decide whether to exit early. This condition uses a confidence score computed from the model prediction and a threshold set in advance which needs to be satisfied. Algorithm 1 shows the implementation of the cascading method we found to be most effective.

---

**Algorithm 1** Early exit model cascade with maximum softmax confidence metric and no ensembling

---

**Require:** input tensor $\mathbf{X}$, models $\{M_1, ..., M_n\}$, thresholds $\{t_1, ..., t_{n-1}\}$, $n \geq 2$, $n \in \mathbb{N}$
    **for** $i = 1, ..., n$ **do**
        $\boldsymbol{z}_i = M_i(\mathbf{X})$
        $\boldsymbol{p}_i = \text{softmax}(\boldsymbol{z}_i)$
        **if** ($i < n$ **and** $\max(\boldsymbol{p}_i) \geq t_i$) **or** $i == n$ **then**
            return $\text{argmax}(\boldsymbol{p}_i)$                 ▷ cascade returns predicted class

---

**Cost**    Let $C_i$ be the cost of a model $M_i$ which represents metrics we want to improve, such as number of Multiply-Accumulate (MAC) operations or time of inference. We order the models within a cascade so that $C_i < C_{i+1}$, meaning the cheaper model makes a prediction first. When there are $n$ models in the cascade, there are $n$ possible costs when inferring a single input. In the case $n = 2$, when an early exit is made, the cost is $C_1$, otherwise the cost is $C_1 + C_2$. They represent the lower and upper bound for the cascade. Assume we exit early after the first model at a rate $\varepsilon_1$, the cost becomes $C_1 + (1 - \varepsilon_1)C_2$ in average. For cost reduction, we want to achieve $C_1 + (1 - \varepsilon_1)C_2 < C_2$. Therefore, our early exit rate must satisfy $\varepsilon_1 > C_1/C_2$. Assume we want to maintain the second model's accuracy $A_2$. For a two-model cascade, the improvement factor $I$ by which we reduce the cost is $I = C_2/(C_1 + (1 - \varepsilon)C_2)$. Therefore, the improvement we can achieve is limited by the early exit rate we can reach before cascade accuracy declines and the cost difference between the models, where larger is better for both.

**Early exit condition**  We need a condition to decide when to exit early. Such a condition is typically evaluated by obtaining a confidence score from the model prediction and comparing it to a threshold $t$. This is done after each model in the cascade except the last, resulting in $n-1$ thresholds. For classification tasks, the model output for an input tensor **X** is usually a logits vector $z$ and the predicted class is $\text{argmax}(z)$. We can use the softmax function

$$\text{softmax}(z_i) = \frac{e^{z_i}}{\sum_{j=1}^{K} e^{z_j}} = p_i \qquad (1)$$

to transform the logits into pseudo-probabilities $p$ for all $K$ classes. One confidence metric is the maximum softmax, which we can use as probability that the prediction is correct. We exit early if $\max(p) \geq t$. Various other conditions exist in literature (Han et al., 2021; Xu & McAuley, 2022) and the entropy threshold appears to be most common. After applying the softmax function we can compute the Shannon entropy

$$H(p) = -\sum_{i=1}^{K} p_i \log(p_i). \qquad (2)$$

A lower entropy represents higher confidence in the prediction and we exit early when $H(p) \leq t$. Other confidence metrics found in literature are the logits margin (Streeter, 2018) and the softmax margin (Park et al., 2015), which are the largest logit or softmax minus the second largest respectively. They represent by how much the model is more confident in its first prediction and we exit if the margin is above or equal to a threshold.

**Ensemble**  When the cascade does not exit after the first model, we have multiple predictions which we can ensemble in the hope of improving performance. Typical ways to ensemble models are plurality vote (Hansen & Salamon, 1990) or averaging of the outputs Bishop et al. (1995). To mean ensemble the outputs, we compute the arithmetic mean of softmax or logits. An intuitive alternative is to compare the prediction confidence of all models and choose the most confident, which we call comparison ensemble. Weighting or calibration can be used when model confidences are mismatched so that the most overconfident model does not dominate the ensemble. Guo et al. (2017) demonstrate that temperature scaling is an effective calibration tool and Ashukha et al. (2020) recommend it for ensembles. Temperature scaling is achieved by dividing the logits by a temperature factor $T$ so that the calibrated probabilities become

$$\hat{p}_i = \frac{e^{\frac{z_i}{T}}}{\sum_{j=1}^{K} e^{\frac{z_j}{T}}} \qquad (3)$$

where $T$ is set to minimize the negative log-likelihood.

## 4 EXPERIMENTS

In order to evaluate the impact of early exit model cascades, we verify whether they reliably outperform state-of-the-art single models across entire Pareto fronts on different tasks. We rigorously compare many different cascading methods to determine the most effective. To understand when cascades work well, we figure out how properties such as size and accuracy of cascaded models affect performance. We begin by conducting a set of experiments on ImageNet (Russakovsky et al., 2015) (Section 4.1) due to its relevance in related work. As a step to indicate general utility of model cascades, we demonstrate Pareto improvement on text classification tasks from the GLUE (Wang et al., 2018a) benchmark (Section 4.2). All pretrained models used for image and text classification are publicly available from PyTorch Image Models (Wightman, 2019) and Hugging Face (Wolf et al., 2020) respectively. We use a GeForce RTX 3090 GPU for all inference. The code to reproduce these experiments is available in the supplementary material.

### 4.1 IMAGE CLASSIFICATION ON IMAGENET

We select from 668 pretrained models with tabulated accuracy for the ImageNet validation set from PyTorch Image Models (Wightman, 2019). For the computational cost of models we obtain MAC with the fvcore library. To acquire the time cost we use benchmark numbers in inferred samples per

second on an RTX 3090 with NHWC data format and automatic mixed precision from Wightman (2019). From this we can establish baseline Pareto fronts for both MAC and time cost. We infer the ImageNet validation set for models at the Pareto fronts to confirm their accuracy and obtain logits, from which we compute prediction correctness and confidence score for every image. From these we can efficiently compute exact accuracy, threshold and average inference cost at all possible early exit rates for every cascade. Threshold selection can then be made based on desired trade-off. For example we choose the threshold at the point of maximal validation Pareto improvement for our ImageNet test server submissions.

As a strong and continuous baseline we use the Pareto front established by linear interpolation between pretrained model pairs, which represents the trade-off achieved when randomly selecting one of the two models with varying probability. No linear interpolation is done between points at cascade Pareto fronts to have a direct comparison between single cascades and random model selection. We evaluate improvement by cascade Pareto fronts as a factor by which average inference cost in MAC or time is reduced to achieve the same accuracy. The average cost represents the total cost for the entire validation set at specific thresholds divided by number of images.

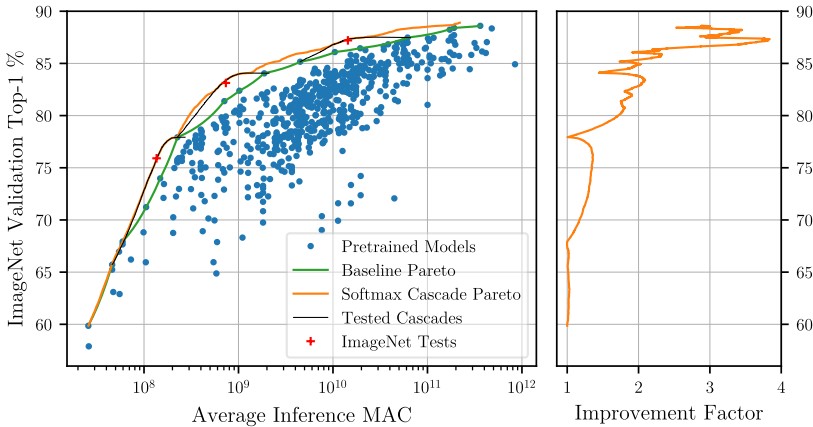

Figure 1: The maximum softmax confidence metric without calibration and ensembling is one of the simplest yet most effective for cascading and achieves Pareto improvement across most of the MAC range with 2-model cascades. The pretrained models show that the accuracy benefit from increasing model size diminishes. Cascade performance improves as accuracy difference between models decreases compared to MAC difference. Two points at the cascade Pareto front with locally maximal improvement were evaluated on the ImageNet test set. The results are summarized in Table 1 and confirm the validation results. Notable are that cascading also increases the top accuracy by 0.306% to 88.90% and that the improvement factor drops to almost 1 at 77.92% accuracy due to the outlying MobileNetV3-Large pretrained on ImageNet-22K (Ridnik et al., 2021).

Table 1: Two points at locally maximal improvement from the softmax 2-model cascade Pareto front were evaluated on the ImageNet test set as shown in Figure 1. They were chosen as strongest points at the cascade Pareto front to test against validation set overfitting. Each point corresponds to a cascade with a specific threshold. Values are rounded to 4 significant digits.

| Cascade Models | | Validation | | Test | | Softmax |
| --- | --- | --- | --- | --- | --- | --- |
| First | Second | Top-1 ↑ | MAC ↓ | Top-1 ↑ | MAC ↓ | Threshold |
| MN3-S 0.75 (Howard et al., 2019) | MN3-L (Ridnik et al., 2021) | 75.88% | 136.2M | **75.91%** | **135.9M** | 0.4551 |
| MN3-L Ridnik et al. (2021) | EN B3 NS Xie et al. (2020) | **83.39%** | **732.4M** | 83.13% | 735.2M | 0.7112 |
| EN B4 NS (Xie et al., 2020) | BEiT$_{224}$-L (Bao et al., 2021) | **87.32%** | 14.39G | 87.21% | **14.38G** | 0.5658 |

For the trade-off between accuracy and MAC, the baseline Pareto front consists of 27 models. For 2-model cascades, there are $\binom{27}{2} = 351$ possible combinations. Figure 1 shows the Pareto front and improvement factor for the cascading method which uses maximum softmax confidence without calibration or ensembling. The cascades achieve improvement along most of the Pareto front with a reduction in computational cost by a factor of up to 3.83 for the same accuracy. This is achieved

by the cascade containing the models EfficientNet B4 NS (Tan & Le, 2019), a convolutional neural network pretrained on 300M unlabeled images from the JFT dataset (Hinton et al., 2015; Chollet, 2017), and BEiT$_{224}$-L (Bao et al., 2021), a vision transformer pretrained on ImageNet-22K, which shows the benefit of model diversity. Model families which contribute to the cascade Pareto front in descending order of importance are EfficientNet NoisyStudent (Xie et al., 2020), MobileNetV3 (Howard et al., 2019; Ridnik et al., 2021), BEiT (Bao et al., 2021), TinyNet (Han et al., 2020b), GhostNet (Han et al., 2020a) and LeViT (Graham et al., 2021).

To test against validation set overfitting, points of the cascade Pareto front with locally maximal improvement were evaluated on the ImageNet test set. As shown in Table 1, test set performance is close enough to confirm the validation set results. However, test accuracy is slightly lower in average. Possible explanations are that the optimal points are shifted for the test set, that the cascade Pareto front envelops noise and that the pretrained models themselves are slightly overfitted to the validation set. To investigate the last point, the pretrained EfficientNet B4 NoisyStudent was evaluated on the ImageNet test set and achieved a top-1 accuracy 0.04% lower than on the validation set, which accounts for almost half of the 0.09% difference for the 3rd test point.

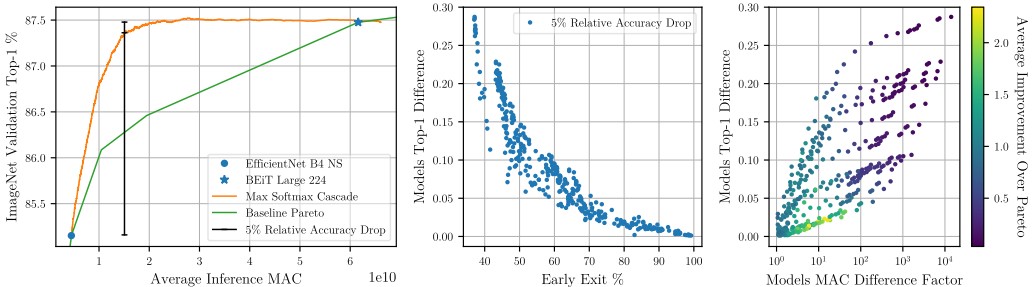

Figure 2: **Left:** The cascades tend to maintain the accuracy of the better model for a plateau before accuracy starts to break down. **Middle:** Each point represents the largest early exit rate before cascade accuracy drops by 5% of the top-1 difference below the better model. **Right:** The average Pareto improvement of all individual cascades. While a lower accuracy difference between cascaded models is always desirable, there exists a sweet spot for the size difference. While the length of the plateau before accuracy drops increases with smaller accuracy difference between models, for the baseline Pareto this plateau grows with the size range as the accuracy benefit of increased model size diminishes.

Figure 2 shows the relationship between the accuracy difference of the models in the cascade and the early exit rate cascades can achieve before accuracy starts to break down. The cascades perform better when the accuracy difference between models is small while the MAC difference is large. Therefore, the Pareto improvement is minimal in the lowest MAC regime and becomes larger at higher MAC regimes because of the diminishing accuracy benefit from increased model size, which can be observed in Figure 1.

**Cascading method comparison** We compare the 4 confidence metrics entropy, maximum softmax, softmax margin and logits margin both without ensembling and with comparison ensembling enabled by calibration using temperature scaling. Furthermore, using the maximum softmax confidence metric we evaluate mean ensembling for both logits and softmax, with and without temperature scaling. This results in 12 different cascading methods compared in Figure 3.

From the methods without ensembling, the entropy confidence metric performs worst. Entropy is influenced by probabilities assigned to false labels while we are only interested in whether the predicted label is correct for early exiting. Softmax margin is the only cascading method which achieves an improvement >1 across the entire Pareto front (except at the smallest model, whose computational cost cannot be reduced further with cascading). It also reaches the largest average Pareto improvement of all 12 methods with ×1.502 across the entire accuracy range. This is because softmax margin dominates below 78% accuracy, which represents almost 2/3 of the accuracy range but only about 1/4 of the log scaled MAC range. The maximum softmax metric has the second largest average improvement at ×1.497 and shows most MAC range dominance of all methods.

When ensembling, calibration may be required, because otherwise there is no guarantee that the uncalibrated confidence scores of the different models are comparable. Temperature scaling was done on the validation set, which means that the results for methods using temperature scaling represent an ideal scenario where the calibration is optimally fitted to the data. Yet, calibration proves ineffective for mean ensemble cascades and the comparison ensemble methods achieve the least improvement of all. Mean ensembling shows potential at the upper end of the accuracy range. Therefore, we use the maximum softmax confidence metric, which is worse overall but better at higher accuracies than the softmax margin (see Appendix C for details). While the mean softmax method performs better overall, uncalibrated mean logits ensembling achieves the best average improvement above 86% accuracy of all cascading methods, as well as the best top accuracy of 89.03%. A possible explanation for the poor performance of ensembling is that Pareto optimal cascade settings rely on high early exit rates, where only the least certain predictions are forwarded. These are less useful for or may even hurt ensembling.

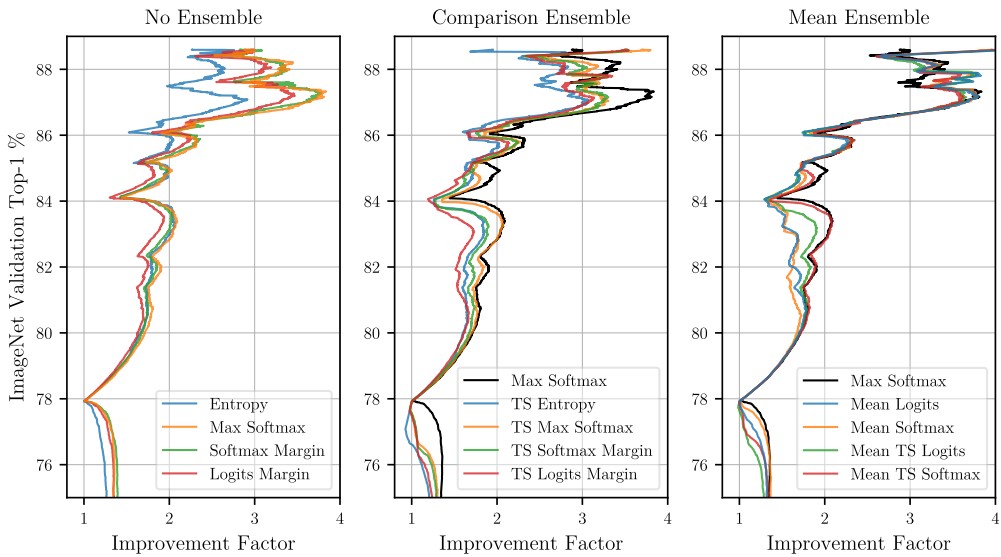

Figure 3: A comparison of Pareto improvement achieved for the 12 tested cascading methods. To improve legibility, only the accuracy range above 75% is shown as improvement achieved below that point is small for all methods. The maximum softmax method from the left is plotted again as black line to have a baseline for comparison. TS stands for temperature scaled.

**3-model cascades** Cascading more models increases the complexity since well suited models are required for improvement and multiple thresholds need to be selected, which increases the problem dimensionality. The ImageNet validation set consists of 50000 images, which means that there are 50001 choices per threshold ranging from always to never exiting early. For the 27 Pareto optimal models there are $\binom{27}{3} = 2925$ possible combinations. Since computing cascade performance is cheap, exhaustive search is still feasible for 3-model cascades and has the advantages of simplicity and a guarantee that the optimum will be found. To do so, for every model combination we set the first threshold and then compute the cascade performance for all choices of the second threshold. We can speed up the process by only sampling every tenth sorted value for the first threshold since neighboring values are almost identical.

Interestingly, ensembling performs slightly worse relative to no ensembling for 3 instead of 2 models and mean softmax is now superior to mean logits. The maximum softmax method achieves the largest average Pareto improvement of all cascading methods both across the entire accuracy range and above 80% only. Figure 4 shows that there is diminished benefit from adding a third model but improvement at higher accuracies. 2-model cascades offer a strong and practical baseline, but for optimal performance adding more models should be considered while keeping in mind the trade-off between diminishing returns and increased complexity, which could also harm generalizability. To test this, we evaluated the point of the 3-model maximum softmax Pareto front with largest improvement on the ImageNet test set. This is the cascade with models EfficientNet B4 NS (Tan &

Le, 2019), BEiT$_{224}$-L (Bao et al., 2021) and EfficientNet L2 NS at the thresholds $t_1 = 0.6299$ and $t_2 = 0.4905$. While test accuracy at 87.86% is only 0.08% lower than on the validation set, average MAC is also 1.1% larger at 26.32G. This indicates slightly worse generalization than the 3rd test result in Table 1 for a cascade with only the first 2 models.

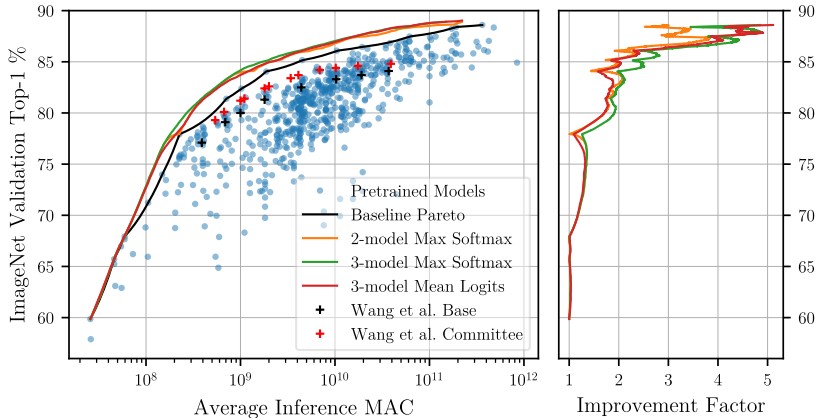

Figure 4: 3-model cascades start to outperform 2-model cascades at higher MAC regimes. **Left:** We compare our results to Wang et al. (2022), who use the mean logits ensemble method. **Right:** The maximum softmax method achieves the best performance of all cascading methods and outperforms the mean logits method used by Wang et al. (2022) across most of the Pareto front.

**Inference time** Radosavovic et al. (2020) show that the number of MAC operations is an imperfect predictor for GPU inference speed. Therefore, we also evaluate cascade performance for the trade-off between accuracy and speed measured in average time per inference. Figure 5 shows that 2-model cascades already achieve improvement everywhere except near the outlying LeViT-128S (Graham et al., 2021). As can be seen in greater detail in Appendix A, maximum softmax again performs best of all tested cascading methods, which demonstrates its reliability and consistency. Notably, it outperforms mean ensemble methods even above 86% accuracy for inference speed, achieving an average improvement of ×2.618 compared to ×2.203 by the mean logits cascading method. The baseline Pareto front consists of 25 models. There is a large model overlap between MAC and speed Pareto fronts. The biggest difference is that EfficientNet Tan & Le (2019); Xie et al. (2020) and the related TinyNet Han et al. (2020b) architecture families perform worse for inference speed while LeViT Graham et al. (2021) and ConvNeXt Liu et al. (2022) perform better.

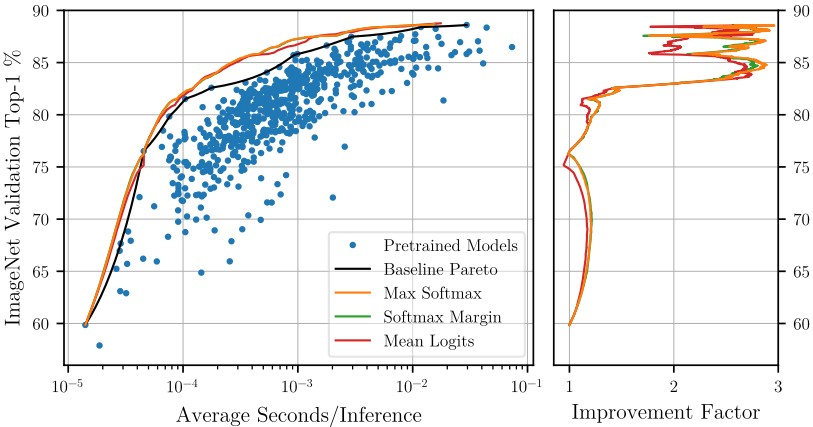

Figure 5: Inference speed is measured in average time per image on an RTX 3090 GPU. The images are batched, therefore the actual latency is larger. Less Pareto improvement is achieved than for MAC because the time cost spans 4 orders of magnitude instead of 5. The 2-model cascades perform particularly well at concave points of the baseline Pareto front.

## 4.2 TEXT CLASSIFICATION ON GLUE BENCHMARK

Early exit model cascades are effective for image classification with both convolutional neural networks and vision transformers. We expect that the cascades also work in other domains where prediction confidence can be quantified. Indeed, beneficial early exiting has been demonstrated for image segmentation (Wang et al., 2022) and reinforcement learning (Wołczyk et al., 2021). As a first step to test that the identical cascade structure can be used in a different domain, we experiment on the text classification tasks SST-2 (Socher et al., 2013), MRPC (Dolan & Brockett, 2005), QNLI (Rajpurkar et al., 2016) and QQP (Chen et al., 2018) from the GLUE (Wang et al., 2018a) benchmark. The results are shown in Figure 6 and confirm the efficacy of early exit cascades. Since the tasks are binary classification, the 4 confidence metrics entropy, maximum softmax, softmax margin and logits margin result in identical order when sorting the confidence scores of all predictions, and the resulting cascades are equivalent. We searched for fine-tuned models on Hugging Face (Wolf et al., 2020). Models used include BERT (Devlin et al., 2018), RoBERTa (Liu et al., 2019), DeBERTa (He et al., 2020), ELECTRA (Clark et al., 2020), ALBERT (Lan et al., 2019), DynaBERT (Hou et al., 2020), DistilBERT (Sanh et al., 2019), M-FAC (Frantar et al., 2021), MiniLM (Wang et al., 2020) and XLNet (Yang et al., 2019).

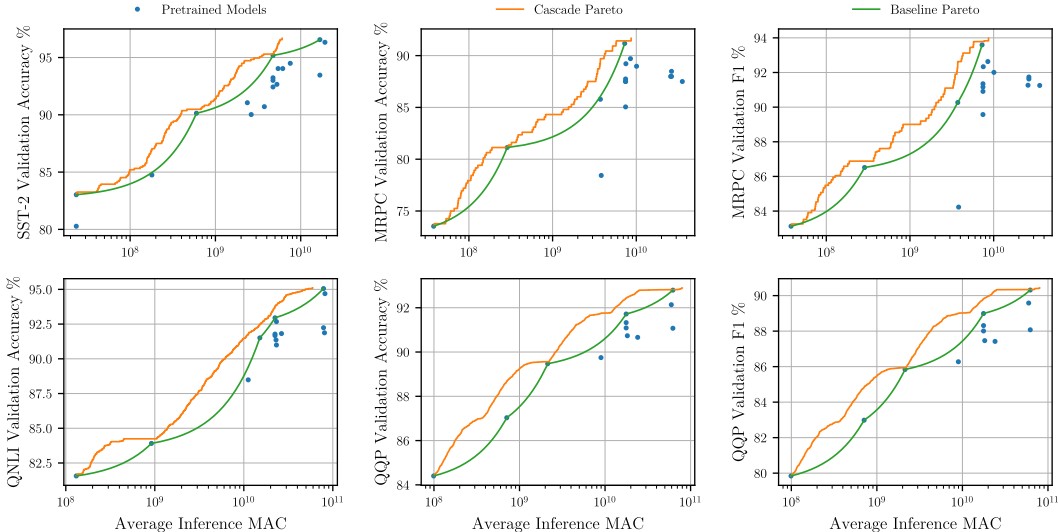

Figure 6: Early exit cascades achieve Pareto improvement for text classification tasks from the GLUE benchmark in spite of the large accuracy differences between models. The results for the MRPC and QQP tasks are similar for both accuracy and F1 score.

## 5 CONCLUSION

We demonstrate the efficacy of early exit cascades on a wide selection of publicly available pretrained models for image classification on ImageNet (Russakovsky et al., 2015) and text classification tasks from the GLUE benchmark (Wang et al., 2018a). We compare various early exit conditions, ensembling strategies and temperature scaled calibration. Notably we find that the commonly used entropy threshold performs worst, that ensembling can increase the maximum accuracy but fails to improve the Pareto front and that temperature scaling is ineffective. Which method performs best varies based on the cascaded models. Since comparing cascade performance of different methods is computationally cheap and fast, the best situational method can be determined. Otherwise, our recommendation is to use the maximum softmax threshold condition without ensembling and calibration, which is not just simple and reliable but also the most effective of all methods we tested.

REPRODUCIBILITY STATEMENT

We provide code to reproduce our experiments in the supplementary material. Our code can be executed on regular consumer hardware in a few GPU and CPU hours except for the 3-model cascade exhaustive search, which may take up to a few CPU days. For the GLUE benchmark experiments we also provide all data generated by the submitted code, which includes stored Pareto fronts, logits, inference data, dataframes with model information and more. For the ImageNet experiments we provide all Pareto fronts but do not include the stored cascades, logits and inference data due to their large file size. No training was done and all datasets and pretrained models used in this work are publicly available. Therefore, we provide a strong and reproducible baseline for future related research that is currently missing.

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

# A  ADDITIONAL PARETO IMPROVEMENT COMPARISON DATA

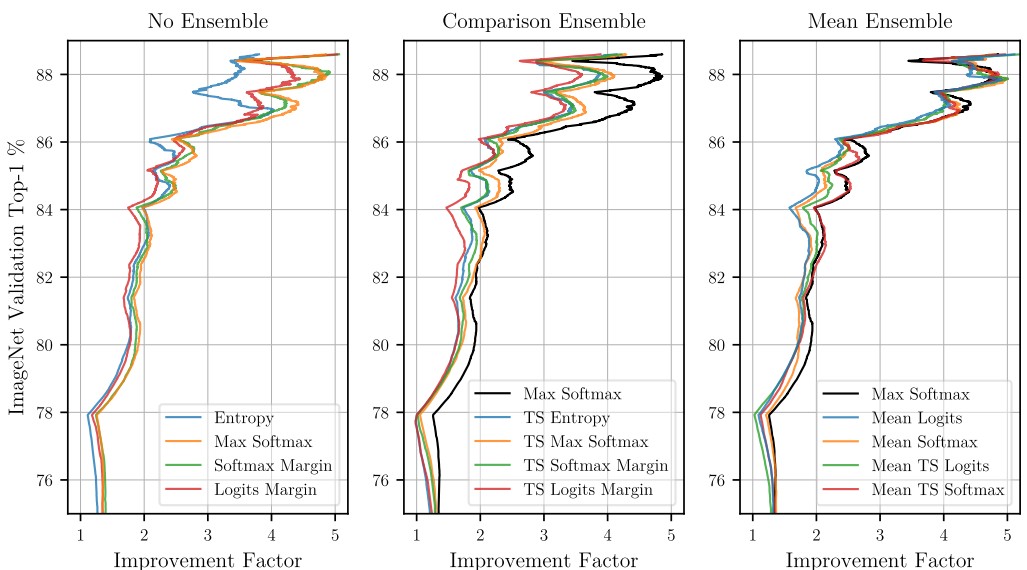

Figure 7: 3-model cascade Pareto improvement for the accuracy-MAC trade-off. Maximum softmax confidence without ensembling is the best cascading method overall and is plotted again in the middle and right as black line to have a baseline for comparison. The softmax margin cascading method performs best in the lower accuracy range and mean ensembling shines at the upper end of the accuracy range.

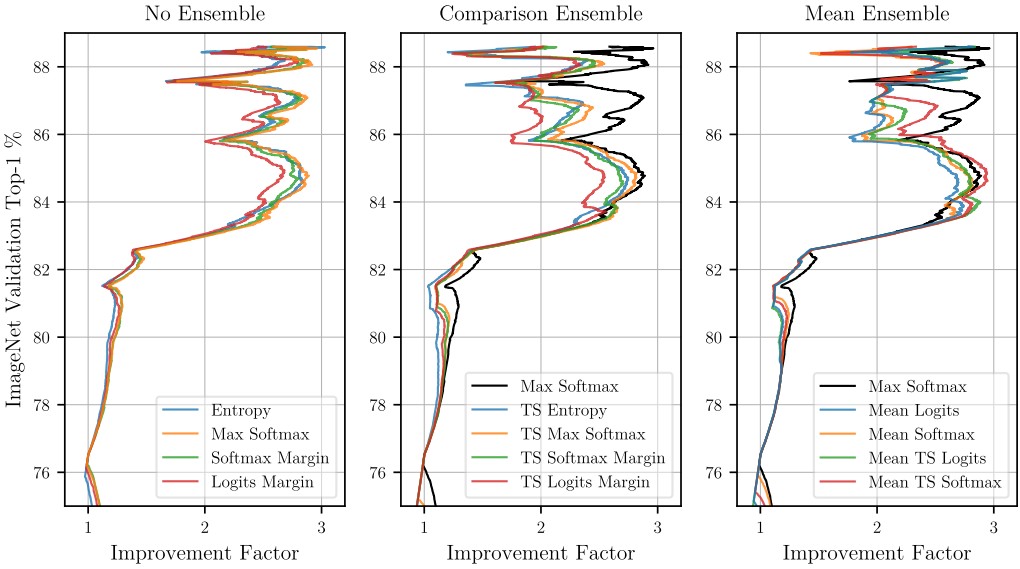

Figure 8: 2-model cascade Pareto improvement for the accuracy-time trade-off. Again, maximum softmax confidence without ensembling is the best cascading method of all and is plotted again in the middle and right as black line to have a baseline for comparison. Notable is that mean ensembling shows worse performance for inference speed relative to the maximum softmax method. Furthermore, the mean softmax method now outperforms the mean logits method. This indicates that cascading methods which use ensembling are less reliable.

## B    TOP-5 ACCURACY

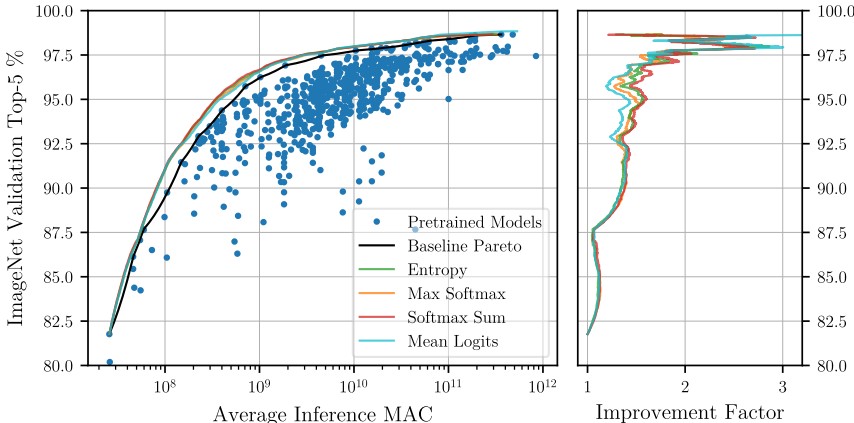

Figure 9: For some applications, such as content or autocomplete recommendation, it is less important that the top-1 prediction is correct and instead it matters more that the correct prediction is among the top. To investigate this scenario, we evaluate the trade-off between top-5 accuracy and MAC. 2-model cascades perform better at the lower range but worse at the higher range for top-5 accuracy in spite of the smaller accuracy differences. Here, the entropy confidence metric beats maximum softmax since it takes into account more than the top-1 prediction. However, the top-5 softmax sum, which takes into account all relevant probabilities, outperforms entropy across most of the Pareto front. In an attempt to improve the cascading further, we try mean ensembling. However, neither the mean logits nor the mean softmax (not pictured) method perform well outside of narrow points at the upper limit of the accuracy range.

## C    CONFIDENCE METRIC FOR MEAN ENSEMBLES

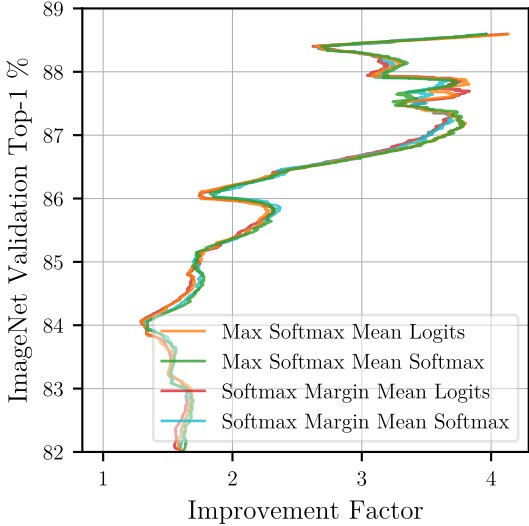

Figure 10: A comparison of confidence metrics for mean ensembled 2-model cascades in the high accuracy regime, where they perform best. While the difference is very small, the softmax margin confidence metric is better overall than maximum softmax for mean ensembled cascades. However, maximum softmax confidence leads to better improvement in the high accuracy regime, where mean ensembles perform best, and is therefore the metric we selected.

