# OpenReview forum: "Conditional Execution Of Cascaded Models Improves The Accuracy-Efficiency Trade-Off"
_ICLR.cc/2023/Conference — Submitted to ICLR 2023_

### Official Review · Reviewer_Bbfc · 2022-10-23

**Confidence:** 3
**Correctness:** 4
**Technical Novelty And Significance:** 1
**Empirical Novelty And Significance:** 2
**Recommendation:** 3

**Clarity, Quality, Novelty And Reproducibility:**

The paper is clearly structured and well written. It appears the result should be reproducible.

**Strength And Weaknesses:**

Strength:
* A comprehensive and careful study of some key design decisions of cascade models.
* Extended evaluation to the text domain.

Weakness:
* Novelty is a bit limited: the paper investigated some common obvious choices (in terms of choice of confidence metric and calibration procedure) and provided some empirically-supported conclusions. Otherwise, the cascade model itself is not too different from previous work (e.g., [Wang et al. 2022](https://openreview.net/pdf?id=MvO2t0vbs4-)).

**Summary Of The Paper:**

This paper conducts empirical investigation regarding the best practice of building cascade models. Specifically, the paper investigated the impact of early-exist condition and the choice of model calibration on the efficiency-accuracy performance of the cascade, and investigated the performance of both vision and text models. The experiments reveals that the maximum softmax achieves most improvement overall, and temperature scaled calibration hurts performance.

**Summary Of The Review:**

As mentioned in the weakness, the paper investigated some common obvious choices (in terms of choice of confidence metric and calibration procedure) and provided some empirically-supported conclusions. Otherwise, the cascade model setup is not too different from previous work (e.g., [Wang et al. 2022](https://openreview.net/pdf?id=MvO2t0vbs4-)).
While it is useful to share empirical results, I feel the submission in its current form is sufficient as a high-quality workshop submission, but does not carry sufficient scientific significance to justify a publication at top venue like ICLR.

---

> ### Author Response · Authors · 2022-11-18
> **Response to Reviewer Bbfc**
>
> Thank you for taking the time to review our submission!
>
>
>
> >Novelty is a bit limited: the paper investigated some common obvious choices (in terms of choice of confidence metric and calibration procedure) and provided some empirically-supported conclusions. Otherwise, the cascade model itself is not too different from previous work (e.g., Wang et al. 2022).
>
> We have clarified our contributions and added a comparison to Wang et al. (2022) in Figure 4. We list the following key contributions in our revision:
>
> 1. We demonstrate that already 2-model cascades dominate the entire ImageNet Pareto front.
>
> 2. We provide insight into how to construct the model cascades: We empirically demonstrate a relationship between accuracy difference of cascaded models and achievable early exit rate as well as desirable size difference in Figure 2. With this we explain why cascades perform better in the large model domain on ImageNet.
>
> 3. We provide a nuanced answer to which confidence metric is best. Overall, maximum softmax achieves the most improvement. The commonly used entropy threshold performs worst.
>
> 4. We find that using only the latest prediction often performs better than ensembling all the predictions and that temperature scaled calibration is ineffective at alleviating this.
>
> 5. For our evaluation we rely on an external baseline comprised of Pareto optimal public models to ensure our results are credible and relevant. This means we provide a strong and reproducible baseline for future related research that is currently missing. A strong evaluation baseline is valuable because the choice of baseline has a big impact on the measured performance of a method.
>
>
>
> We appreciate all feedback and will gladly address any further issues or concerns!

---

### Official Review · Reviewer_J6Fm · 2022-10-24

**Confidence:** 4
**Correctness:** 3
**Technical Novelty And Significance:** 1
**Empirical Novelty And Significance:** 1
**Recommendation:** 3

**Clarity, Quality, Novelty And Reproducibility:**

The results are reproducible due to the simplicity of model cascades. The clarity needs to be improved, particularly what’s the motivation of conducting these experiments and what new knowledge the community could benefit from this work needs to be made much clearer (considering the existing results in [A]).


**Strength And Weaknesses:**

The results presented in this work largely make sense to me. But a major concern is that the difference between this work and previous work [A] is unclear. Many conclusions presented in this work were already verified in [A] except that [A] focuses on image classification while this work provides both image & text classification results.

For example, the first contribution “2-model cascades dominate the accuracy-compute Pareto front” was shown in Figure 7 in [A]. The comparison of different confidence metrics (entropy, margin, etc) is similar to Figure 3 in [A].

Therefore, I am unsure about the new value this work could add to the existing knowledge about model cascades.

[A] Wisdom of Committees: An Overlooked Approach To Faster and More Accurate Models. ICLR 2022.


**Summary Of The Paper:**

This work explores model cascades as a way to improve the accuracy-efficiency trade-off of deep neural networks. They conduct experiments to evaluate different design choices in model cascades, such as the confidence metric, number of models in the cascade, and whether to calibrate the logits before ensembling. Experimental results on image classification and text classification are provided to support the analysis of model cascades.


**Summary Of The Review:**

This work does not propose new techniques but rather analyze the various design choices of an existing method (model cascades). This is fine as long as the analysis results are informative. However, considering a very similar prior work [A], the contribution of this work is unclear and seems to be modest. So I vote for rejection.

---

> ### Author Response · Authors · 2022-11-18
> **Response to Reviewer J6Fm**
>
> Thank you for the valuable comments, we appreciate it!
>
>
>
> >The results presented in this work largely make sense to me. But a major concern is that the difference between this work and previous work [A] is unclear. Many conclusions presented in this work were already verified in [A] except that [A] focuses on image classification while this work provides both image & text classification results.
>
> We agree that [A] is highly relevant prior work. We have made our contributions clearer in order to address your concern. Specifically, we demonstrate that early exit model cascades improve over state-of-the-art single models across the entire ImageNet Pareto front already in the 2-model setting. In addition, [A] consider only *committees*, not cascades made up of different architectures.
>
> Furthermore, we provide insight into how to construct the model cascades. We empirically demonstrate a relationship between accuracy difference of cascaded models and achievable early exit rate as well as desirable size difference in the plots on the middle and right of Figure 2. We provide a nuanced answer to which confidence metric is best. Notably, we find that using only the latest prediction generally performs better than ensembling all the predictions (as is done in [A]) and that temperature scaled calibration is ineffective at alleviating this. For our evaluation we rely on an external baseline comprised of Pareto optimal public models to ensure our results are credible and relevant. This means we provide a strong and reproducible baseline for future related research that is currently missing. A strong evaluation baseline is valuable because the choice of baseline has a big impact on the measured performance of a method.
>
>
>
> >For example, the first contribution “2-model cascades dominate the accuracy-compute Pareto front” was shown in Figure 7 in [A].
>
> Our Figure 4 is a direct comparison for 3-model cascades between the method used by [A] (mean logits ensemble) and the method we find to be best (maximum softmax confidence without ensembling). We have added the results reported by [A] to Figure 4 as well as clarified the caption to demonstrate that our baseline is stronger and our recommended method performs better. We demonstrate improvement across a continuous Pareto front compared to a wide range of state-of-the-art models and do so for multiple methods and diverse models instead of a committee. [A] demonstrate improvement over their own single models but not over the single model Pareto front.
>
>
>
> >The comparison of different confidence metrics (entropy, margin, etc) is similar to Figure 3 in [A].
>
> Figure 3 in [A] shows a common method to compare confidence metrics for a single model. However, in our experiments we found this method to be lacking. Often, the best metric for a single model does not align with the best metric for a cascade. We clarify in our revision that we evaluate different metrics and cascading methods across entire Pareto fronts in multiple settings using many different models to obtain a thorough and reliable comparison.
>
>
>
> >The clarity needs to be improved, particularly what’s the motivation of conducting these experiments and what new knowledge the community could benefit from this work needs to be made much clearer (considering the existing results in [A]).
>
> To address this, we improved the clarity of our motivation for the experiments in the beginning of the experiments section as well as the contributions in the introduction.
>
> Some key motivations for conducting these experiments are:
>
> 1. Verify that model cascades reliably outperform state-of-the-art single models across entire Pareto fronts on different tasks.
>
> 2. Rigorously compare many different cascading methods to determine the most effective.
>
> 3. Figure out how properties such as computational cost and accuracy of cascaded models affect performance.
>
> 4. Provide a reproducible and strong baseline for the community.
>
>
>
> If any issues, concerns or questions remain, please let us know and we will gladly address them!

---

> ### Comment · Reviewer_J6Fm · 2022-11-30
> **Concerns remain after rebuttal**
>
> Thanks for providing the detailed response.
>
> After reading the rebuttal, I still feel the difference between this work and [A] is unclear.
>
> > Specifically, we demonstrate that early exit model cascades improve over state-of-the-art single models across the entire ImageNet Pareto front already in the 2-model setting.
>
> [A] also demonstrates strong results in 2-model cascades, e.g., Figure 4 & 7 in their paper.
>
> > [A] consider only committees, not cascades made up of different architectures
>
> According to [A], they also use different architectures in cascades (e.g., Table 16), although they didn't use different architecture families.
>
> There are some differences in the details between this work and [A]. But those differences are not enough to justify the contribution of a top-conference paper. Therefore, I keep my original rating.

---

### Official Review · Reviewer_CNbJ · 2022-10-25

**Confidence:** 4
**Correctness:** 4
**Technical Novelty And Significance:** 4
**Empirical Novelty And Significance:** 4
**Recommendation:** 8

**Clarity, Quality, Novelty And Reproducibility:**

The paper is well written and their experimental methodology is high quality. I expect their results will be easily reproducible given the use of publicly available models and data. I think the novelty of the study is high and the results are potentially impactful.

**Strength And Weaknesses:**

I found the paper to be clear, focused and interesting. One strength of the study is their experiment design - in particular, taking advantage of the large number of pre-trained models available through frameworks like TIMM and HuggingFace to answer questions about cascading methods convincingly. The insights from the study are also impactful, with 2-5x reductions in computation and up to 3x reductions in measured inference time.

One question I had was about the precision of the thresholds used. In each of the tables I see four digits of precision used for softmax thresholds. Is this level of precision necessary? What would the results look like if fewer digits are used?

**Summary Of The Paper:**

The authors conduct a thorough investigation of cascading methods with deep neural networks, including threshold tuning, cascades of various depths, heterogenous cascades with different types of architectures and methods of ensembling predictions when early exiting is not possible. The authors demonstrate the efficacy of these techniques for both vision and text tasks.

**Summary Of The Review:**

This paper is well written, well organized and thorough in their analysis of cascading methods for deep neural networks. The results are novel and potentially impactful.

---

> ### Author Response · Authors · 2022-11-18
> **Response to Reviewer CNbJ**
>
> Thank you for the feedback, we appreciate your interest!
>
>
>
> >I found the paper to be clear, focused and interesting. One strength of the study is their experiment design - in particular, taking advantage of the large number of pre-trained models available through frameworks like TIMM and HuggingFace to answer questions about cascading methods convincingly. The insights from the study are also impactful, with 2-5x reductions in computation and up to 3x reductions in measured inference time.
>
> We put a lot of thought into the experiment design and are encouraged that this is being recognized. When measuring a method's efficacy, the choice of the comparison baseline is critical. Using the single model state-of-the-art Pareto front as baseline ensures the results are credible and relevant. In our revision we put more emphasis on our evaluation framework. Since the used models, data and our results are public, we hope to help future related works by providing a strong and reproducible baseline.
>
>
>
> >One question I had was about the precision of the thresholds used. In each of the tables I see four digits of precision used for softmax thresholds. Is this level of precision necessary? What would the results look like if fewer digits are used?
>
> The thresholds which we used for the ImageNet test server submissions were determined automatically at the point of greatest Pareto improvement. The thresholds are therefore exact floats, rounded to 4 significant digits in the table. This level of precision was used out of experimental rigor but is not necessary in practice. If training and application data are well aligned, using too little precision may cause a shift away from optimal improvement. Therefore, we recommend at least 2 digits of precision.
>
>
>
> Please let us know if you have any more questions or feedback. We are happy to address them!

---

### Official Review · Reviewer_Q4Xh · 2022-10-26

**Confidence:** 4
**Correctness:** 3
**Technical Novelty And Significance:** 1
**Empirical Novelty And Significance:** 1
**Recommendation:** 3

**Clarity, Quality, Novelty And Reproducibility:**

The paper is clear, and easy to write. The authors mention that they will release the code, making it reproducible.
For the work that was done, the quality seems good.

**Strength And Weaknesses:**

Strengths:
- The paper is well written
- It is always good to have more results for topics like this, reaffirming what is commonly known through literature.

Weaknesses
- The paper essentially just reconfirms something that is very well known in the conditional computing literature. Early-exiting/cascading helps significantly in models to create a better efficiency trade-off. This has been shown in the past in many studies; although generally at least some part of the network is shared as not to recompute simpler features at the start of the network. Even in relatively new language and vision transformers, many variants of early-exiting have been discussed. For this specific version of early exiting, within a single model, there's already 7 papers out on this topic, showing that it helps to early exit simpler examples.
- This paper really does nothing new on top of what is already known. Although it's good to reconfirm what is known, there is nothing novel in this paper. There are no new insights in this paper worth mentioning.
- The setup is quite archaic, compared to the more commonly used set-up where at least some features are shared, as opposed to using an entirely new network. I don't think it's relevant to show results in this setup anymore, as it is not a very efficient setup to do  early exiting on.
- The fact that temperature thresholding does not work very well has been discussed at length in many early-exiting papers, and things like 'patience' metrics have become more popular for doing early exiting in monolithic networks.



**Summary Of The Paper:**

The paper discusses conditional execution of models from the early-exiting angle; showing that a 2-cascade of models can improve upon baseline performance of executing a single model across the entire pareto front in terms of efficiency and accuracy. The paper takes several combinations of 2 models, and picks good ones, where a threshold is learned to switch from execution from the simple model to a more complex model. The paper also evaluates several techniques for doing the thresholding/cascading.

**Summary Of The Review:**

Reading this paper knowing about conditional computing and early exiting, it does not provide any novel insights. It does reconfirm something that scholars in this area already know, but provides little new insight to the discourse. It has an interesting experimental setup, comparing cascaded ensembles of many existing networks, but this setup will likely not be optimal compared to the more common setup of a monolithic model being early exited from.

All in all, since the paper does not meaningfully contribute to my understanding of early-exiting, nor provides a new method on how to make more efficient networks, I would reject this paper.

---

> ### Author Response · Authors · 2022-11-18
> **Response to Reviewer Q4Xh**
>
> Thank you for taking the time to provide detailed feedback!
>
>
>
> >The paper essentially just reconfirms something that is very well known in the conditional computing literature. Early-exiting/cascading helps significantly in models to create a better efficiency trade-off. This has been shown in the past in many studies; although generally at least some part of the network is shared as not to recompute simpler features at the start of the network. Even in relatively new language and vision transformers, many variants of early-exiting have been discussed. For this specific version of early exiting, within a single model, there's already 7 papers out on this topic, showing that it helps to early exit simpler examples.
>
> >This paper really does nothing new on top of what is already known. Although it's good to reconfirm what is known, there is nothing novel in this paper. There are no new insights in this paper worth mentioning.
>
> We have clarified our contributions in the introduction. Some of the key contributions are:
>
> 1. We demonstrate that already 2-model cascades dominate the entire ImageNet Pareto front.
>
> 2. We provide insight into how to construct the model cascades: We empirically demonstrate a relationship between accuracy difference of cascaded models and achievable early exit rate as well as desirable size difference in Figure 2.
>
> 3. We provide a nuanced answer to which confidence metric is best. Overall, maximum softmax achieves the most improvement. The commonly used entropy threshold performs worst (see Table 3 in *A Survey on Dynamic Neural Networks for Natural Language Processing* for its prevalence in recent works).
>
> 4. We find that using only the latest prediction often performs better than ensembling all the predictions and that temperature scaled calibration is ineffective at alleviating this.
>
> 5. For our evaluation we rely on an external baseline comprised of Pareto optimal public models to ensure our results are credible and relevant. This means we provide a strong and reproducible baseline for future related research that is currently missing. A strong evaluation baseline is valuable because the choice of baseline has a big impact on the measured performance of a method.
>
>
>
> >The setup is quite archaic, compared to the more commonly used set-up where at least some features are shared, as opposed to using an entirely new network. I don't think it's relevant to show results in this setup anymore, as it is not a very efficient setup to do early exiting on.
>
> To address this point, we have clarified paragraphs 3 and 4 in the introduction, where we discuss and compare early exiting from within a single model and early exiting between models in a cascade. They are **two separate and complementary approaches that can be combined**.
>
> To the best of our knowledge, none of the in-model early exiting papers achieve state-of-the-art performance on ImageNet. Feature sharing comes with disadvantages as discussed by Huang et al. (2017). Optimizing early features for early classification hurts performance of later classifiers. The structure of state-of-the-art convolutional neural networks (CNN) is carefully optimized and early classification disrupts this structure, which results in lower accuracy than a separate model at equal computational cost.
>
> As we demonstrate, a smaller accuracy difference between classifiers allows for more frequent early exiting. This motivates using a separate model for early classification, which enables us to increase the early exit rate at the cost of a small overhead when forwarding to the larger model.
>
> Furthermore, we are not limited by the model architecture and **can use pretrained models** for the cascades, which alleviates the training burden and enables finetuning. It also allows us to exploit model diversity. Gontijo-Lopes et al. (2021) show that more diverse models have lower error correlation. Indeed, the best performing 2-model cascade combines a CNN and a Transformer, which are pretrained on different datasets.
>
>
>
> >The fact that temperature thresholding does not work very well has been discussed at length in many early-exiting papers, and things like 'patience' metrics have become more popular for doing early exiting in monolithic networks.
>
> We believe this makes our contribution even more valuable since we demonstrate that already a simple 2-model cascade dominates state-of-the-art Pareto fronts in multiple settings and therefore has great practical utility.
>
> We did not include the patience metric in our comparison since it is not applicable in a setup with at most 2 or 3 classifications.
>
>
>
> If any concerns remain, please let us know and we will be happy to address them!

---

### Decision · Program_Chairs · 2023-01-20

**Decision:**

Reject

**Justification For Why Not Higher Score:**

The reviewers agree that there is no clear distinction from  “Wisdom of Committees: An Overlooked Approach To Faster and More Accurate Models. ICLR 2022.

**Justification For Why Not Lower Score:**

N/A

**Metareview: Summary, Strengths And Weaknesses:**

This work conducts an empirical investigation of various ways to form a cascade from several deep neural networks. These various approaches include thresholding maximum predictive probability, ensembling predictions, and combining models of different architectures. Experiments are performed on both vision and NLP tasks.

The reviewers agree that the submission is well written, and experimental results appear to be reproducible, with sensible experimental setup. The main criticism raised is that all experimental results only reaffirm what is known in the literature. Briefly, these results indicate that ensembling and forming a cascade of models can help to provide a better accuracy and compute trade-off (compared to, say, when one big model is used). The most closely related paper is “Wisdom of Committees: An Overlooked Approach To Faster and More Accurate Models. ICLR 2022.” The reviewers agree that there is no clear distinction from this work, thus limiting novelty. Owing to this main concern, the submission is not ready for publication.

**Summary Of Ac-Reviewer Meeting:**

N/A